# Flexural Performance of Cement-Treated Sand Reinforced with Geogrids for Use as Sub-Bases of Pavement and Railway Structures

**DOI:** 10.3390/ma15082877

**Published:** 2022-04-14

**Authors:** Supphanut Chuenjaidee, Pitthaya Jamsawang, Pornkasem Jongpradist, Xiaobin Chen

**Affiliations:** 1Department of Civil Engineering, King Mongkut’s University of Technology North Bangkok, Bangkok 10800, Thailand; chsupphanut@gmail.com; 2Soil Engineering Research Center, Department of Civil Engineering, King Mongkut’s University of Technology North Bangkok, Bangkok 10800, Thailand; 3Construction Innovations and Future Infrastructures Research Center, Department of Civil Engineering, Faculty of Engineering, King Mongkut’s University of Technology Thonburi, Bangkok 10140, Thailand; pornkasem.jon@kmutt.ac.th; 4Key Laboratory of Heavy-Haul Railway Engineering Structures, School of Civil Engineering, Central South University, Changsha 410017, China; chen_xiaobin@csu.edu.cn

**Keywords:** cement-treated sand, flexural behavior, geogrid, railway, soil stabilization, sub-base

## Abstract

Cement-treated sand (CTS) exhibits undesirable brittle behavior after the applied stress reaches its peak strength. This research investigates the flexural behavior of CTS that is reinforced with uniaxial geogrid (CTSG). A total of 6% cement content was mixed with sand. Uniaxial geogrids with three different strengths were utilized to create the CTSG samples. The number of reinforcement layers, including single and double reinforcements, was studied. The image processing method was applied to analyze the surface cracks in the specimens. The results show that the geogrid type and the number of reinforcement layers affect the flexural behavior of the CTSG. Geogrid reinforcement changed the behavior of the CTS from a brittle material to a semi-brittle or ductile material because the residual tensile stresses were carried by the geogrids. The high-strength geogrid with a double reinforcement layer proved to be most effective in enhancing the peak strength and toughness with improvement ratios of 1.80 and 11.7, respectively. Single and double reinforcement layers with all geogrid types can reduce surface cracks with average crack reduction ratios of 64% and 83%, respectively. The CTSG can be successfully used as a sub-base layer to increase flexural performance and the lifetime of pavement and railway structures.

## 1. Introduction

In civil engineering works, sands have been widely used as filling materials for embankment works and sub-base layers in pavement and railway applications [1,2]. Frequently, sands are mixed with cement to increase the compressive strength and stiffness in the sub-base layer [3,4], called cement-treated sand (CTS) [5,6]. The pavement structures are subjected to tensile and bending stress rather than compressive stress. However, the CTS has significantly lower flexural and tensile strengths than compressive ones. In addition, the CTS shows brittle behavior under flexural stress [7], while ductile behavior is required for the pavement materials to prevent sudden failure from excessive traffic loads [8,9]. An increase in the pavement’s service life is also needed to save maintenance costs [10,11]. Recently, the CTS has been reinforced by randomly distributed fibers to improve the tensile and flexural strengths [1,2]. Fibers redistribute the induced stresses through the tensile stress mobilization of fibers, providing a bridging ability to reduce the brittleness, increase the toughness and reduce the crack caused by the external load [5,12]. However, the homogeneity and uniformity of mixing fiber, cement, and sand cannotbe derived easily in practice [7,10].

Due to its easy and quick installation, geosynthetics are polymeric materials applied to ground improvement works for various civil engineering projects, including road, pavement and airport construction, and railway applications. Recently, geogrids have been used as reinforcement elements in pavement systems, such as stabilizing media in unbound layers, reinforcing elements in asphalt layers, and interlayers in overlay applications [13]. Geogrids are divided into three main types, following aperture shape and rib directions, including uni-axial, bi-axial, and tri-axial. Uni-axial geogrid exhibits high tensile strengths in the uni-directional rib. In contrast, biaxial geogrid ribs have high tensile strength in two directions, and triaxial geogrid with triangular apertures, with the ribs of a tri-axial geogrid oriented in three equilateral directions [14]. The uni-axial geogrid is most suitable for reinforcing the CTSF for sub-base layers because the direction of tensile stresses in the pavement structures induced by the traffic loads is the same as the unidirectional rib of the uniaxial geogrid [13].

Many studies were conducted to investigate the behavior of soil that is reinforced with geogrids through experimental laboratory tests. Kamel [15] studied the effect of geogrid reinforcement on the California bearing ratio (CBR), and the unconfined and triaxial compressive strengths of the subgrade soils. The main results indicated that the optimum position of a single geogrid layer was located at 25% of the sample height from the bottom of the base, sub-base or subgrade layer. Ling and Liu [16] investigate the performances of a flexible pavement reinforced with uniaxial and biaxial geogrid. The geogrid was installed at the interface between the asphaltic concrete and subgrade layers. The results revealed that the geogrid reinforcement could increase the stiffness and strength and reduce the settlement compared to the unreinforced pavement. Virgili et al. [17] reported that the geogrid-reinforced pavement increased resistance to cyclic loads from 66% to 100%. Moayedi et al. [18] investigated the effect of geogrid reinforcement location in paved roads under static loading. The results showed that the geogrid reinforcement placed at the interface between the asphaltic concrete and base layers provided the highest reduction in pavement deflection. However, Al-Azzawi [19] reported that the optimum position of geogrids is at the interface between the base and sub-base layers. Zornberg and Gupta [20] showed that geogrid-reinforced pavements built on clayey soils could minimize crack development, and McCartney et al. [21] reported that geogrid-reinforced soils had high wetting—drying durability based on the field static plate load tests. Singh and Gill [22] performed soaked California bearing ratio tests on subgrade soils with and without geogrid reinforcements. The CBR values for the reinforced specimens were three times higher than those for unreinforced examples. A geogrid placement at 0.2 times the specimen depth yielded the best CBR values.

Most previous studies have focused on using geogrid reinforcement in untreated pavement materials based on the shear strength tests. However, minimal research has been conducted on utilizing geogrids as reinforcement members in the cement-treated sand as a sub-base layer under flexural loading. This research investigates the flexural behavior of cement-treated sand reinforced with geogrid (CTSG) instead of fiber reinforcement. The uniaxial geogrids with three different strengths were utilized to create the CTSG samples. The effect of the number of reinforcement layers was also studied. The flexural performances of CTSGs include flexural stress-deflection characteristics, peak strengths, toughness, failure modes and crack reductions.

## 2. Materials and Methods

### 2.1. Materials

Type I ordinary Portland cement with high calcium oxide content of 63% (Figure 1a) was used as the cementitious agent to develop the unconfined compressive strength of the sand-cement mixtures. The specific gravity and fineness were 3.15 and 2900 cm^2^/g, respectively. The raw sands used in this study (Figure 1b) were derived from Phra Nakhon Si Ayutthaya province, Thailand. This province is a primary source for supplying the natural sand employed for pavement and road materials. General properties were determined following the ASTM standards, as listed in Table 1. Magnification of the sand particles in Figure 1c reveals that most sand particles had angular shapes and rough surfaces. The sand samples consisted of 4% gravel, 95% sand, and 1% fine-grained soils, calculated from the grain size distribution curve in Figure 1d. Due to a narrow particle size range, the sand was classified as poorly graded sand (SP) following the unified soil classification system. The maximum dry unit weight and the optimum moisture content of the sand were 18.4 kN/m^3^ and 13.8%, respectively, obtained from the modified compaction test based on a requirement of the general earthwork standard in Thailand. Three uniaxial geogrid types with different tensile strengths were used in this study, including RE520, RE560, and RE580. The typical feature of uniaxial geogrid in Figure 1e shows the characteristics of aperture shape, dimensions, ribs, and junctions. Table 2 provides the physical and mechanical properties of the uniaxial geogrids that were obtained from the manufacturer.

### 2.2. Specimen Preparation

The cylindrical CTS specimens with a diameter of 50 mm and a height of 100 mm were used for the unconfined compression tests following ASTM D5102 [23] to determine the optimum cement content. The sand samples were oven-dried for 24 h to derive a zero initial moisture content. The 6% cement by dry sand weight showed a 7-day unconfined compressive strength of 1800 kPa, higher than the minimum 7-day strength (1750 kPa) for the cement-treated sub-base material, according to the specification of the Department of Highways of Thailand. In the current study, 6% cement was the optimal cement content utilized for preparing all CTS and CTSG samples. For the flexural strength test, the rectangular CTS and CTSG beam specimens with a width and a height of 100 mm, and a length of 350 mm were used, according to ASTM C1609 [24].

The cement and dry sand were blended in a concrete mixer for 5 min. The sand cement mixture was then mixed with water, following the optimum moisture content of 13.8% for 5 min. Each sample was placed in a steel mold and was subsequently statically compacted in several layers by a hydraulic jack, so that the dry unit weight of each layer reached the specified value of 18.4 kN/m^3^. For the CTSG samples, the number of geogrid reinforcement layers was divided into two types, including a single and a double layer, as shown in Figure 2a. Before placing geogrid in the mold, a strain gauge was installed on the surface rib of the geogrid rib (Figure 1e). Only the lower-layer geogrid was instrumented with the strain gauge at the middle span length to measure the tensile strain developed in a geogrid member due to flexural stress.

The strain gauge was carefully installed and coated to prevent mechanical and moisture damage. A uniaxial geogrid was placed at a level of 20 mm from the bottom of the CTSG sample with the single reinforcement layer. In contrast, two uniaxial geogrids were installed at 20 and 40 mm levels for the CTSG sample with the double reinforcement layer. The sample designation used for flexural tests is listed in Table 3. After completing the molding, the specimen was immediately removed from the mold. The appropriate sample for the flexural test must meet the following criteria: dry unit weight, moisture content, dimension within 2 kN/m^3^, 1%, and 2 mm of the designated values, respectively [5]. The samples passing the criteria were encased with a plastic wrap to prevent moisture content loss [5]. All test specimens were cured for 28 days in the controlled room with a temperature of 25 °C and a humidity of 95% to emulate Thailand’s weather.

### 2.3. Test Method and Measurement Instrumentation

The flexural strength test was performed under the four-point loading condition (Figure 2b), according to ASTM C1609 [24]. The setup and instrumentation of the flexural strength test are shown in Figure 2c. The 50-kN universal testing machine was used to apply the flexural loads under displacement control conditions, as controlled by the electric motor at a constant displacement rate of 0.05 mm/min. A 20-kN load cell was employed to monitor the applied flexural load. The net deflection at the midspan was measured by 20-mm linear variable different transducers (LVDT) installed on the reference beam. The experiment finished when the net deflection reached 2 mm. Enough information points were used to generate the extremely sensitive flexural stress-deflection curves and the flexural load-strain value curves obtained from the data logger.

### 2.4. Parameters Describing Flexural Behavior

The flexural responses of CTSG can be classified as deflection-softening and deflection-hardening [25,26], as presented in Figure 3. The deflection-hardening CTSG provides a higher load-carrying capacity after first cracking than CTS and deflection-softening CTSG. In the current research, the point in the load-deflection curve that clearly illustrates nonlinearity is defined as the first cracking point [25,26]. The flexural stress at this point is termed the first peak strength (*f*_1_), whereas the peak strength (*f*_P_) is defined as the stress at which softening first occurs after *f*_1_. Therefore, *f*_1_ = *f*_P_ is noted in the case of deflection-softening CTSG. A deflection point of *L*/150, corresponding to 2 mm for a 300 mm-span length specimen, is recommended according to ASTM C1609 [24]. The stress carried by CTSG after *f*_1_ at the deflection of *L*/150 is termed the residual strength (*f*_150_). The energy equivalent to the area under the load-deflection curve up to a given deflection is defined as toughness. *T*_150_ is specified as the toughness value at the deflection of *L*/150.

### 2.5. The Image Processing Method

The image processing technique (Figure 4) was used to quantitatively analyze the surface-crack areas of the specimens at the deflection of 2 mm. The crack reduction ratio (*R*_cr_) is defined as the ratio of the surface-crack area of the CTSG sample to the surface crack-area of the CTS sample. The *R*_cr_ is used to study the effect of geogrid reinforcement on the decrease in the surface-crack area compared to the unreinforced sample [27,28]. The surface-crack area is analyzed from the geometric characteristics of the crack pattern. Details about the image processing techniques used to measure crack width and length can be found in the study by Tang et al. [27] and Tang et al. [29].

## 3. Results and Discussion

### 3.1. Characteristics of Flexural Stress—Deflection Curves

Figure 5 shows the effects of the uniaxial geogrid types and the number of reinforcement layers on the flexural stress-deflection responses. The shape of the flexural stress-deflection curves reveals that the uniaxial geogrid types affected the flexural characteristics of the CTSG samples. The stress increases in proportion to the net deflection for the CTS until the first peak stress (*f*_1_) is attained. A sudden decrease in stress is subsequently observed, indicating the deflection-softening response and brittle behavior. Thus, *f*_1_ and *f*_P_ have the same values. Finally, the flexural stress is zero at a large deflection of 2 mm (*L*/150).

Unlike the behavior of the CTSG samples, variations in the flexural stress versus deflection responses can be divided into five stages: (I) linear elasticity stage, (II) brittle failure stage at peak stress, (III) bearing capacity recovery due to the tensile of geogrids, (IV) bearing capacity loss when the cracks propagate upward to the top of beams and (V) residual strength stage. The geogrid reinforcement changed the CTS behavior from a brittle to a semi-brittle or ductile material with less brittleness and higher toughness [3,4,5,30,31,32,33,34,35,36,37,38]. Therefore, CTSG was more effective than CTS in preventing a sudden failure and producing residual strength due to the tensile strength mobilization of the uniaxial geogrids as the deflection continuously occurred [33,34,35]. Moreover, single and double reinforcement layers with all uniaxial geogrid types enhanced the brittle behavior of the CTSG, increased *f*_P_, and delayed the deflection accumulation of the CTS samples [3,4,5,26]. Three categories of flexural stress-deflection curves for CTSG samples were characterized: (1) deflection-softening behavior with *f*_P_ < *f*_150_ (CTSG-520-S), (2) deflection-hardening behavior with *f*_P_ < *f*_150_ (CTSG-560-D and CTSG-580-D), and (3) deflection-hardening behavior with *f*_P_ = *f*_150_ (CTSG-560-S, CTSG-580-S and CTSG-520-D).

### 3.2. The Behavior of Strain Developed in Geogrids

Figure 6 shows the flexural stress versus tensile strain developed in the geogrids. The tensile strain values in the geogrids were induced by flexural stresses and ranged from 0.1 × 10^−3^ to 12 × 10^−3^. The shapes of flexural stress-tensile strain and flexural stress-deflection curves (Figure 5) are similar, and the tensile strains increase with an increase in the net deflection of the specimens. Consequently, the tensile strains were minimal before the *f*_1_ values were reached. After a continuous increase in the net deflections, the first cracks occurred. Then, the tensile strain values became readable, implying that the geogrids began to work and reinforce the CTSG specimens [33]. The strain values continuously increased, following the increasing net deflections, until the strain gauges were damaged. The final strains corresponding to the *f*_P_ values were subsequently recorded. The flexural stress created the shear stress that developed in the interfacial area between geogrid and surrounding sand-cement matrix particle, which damaged the strain gauge at the geogrid surface. Due to the lower elongation property, the high-strength geogrid exhibited a smaller strain level than the low-strength geogrid at the same flexural stress. In the same way, double reinforcement provided a smaller strain value than the single reinforcement, indicating that the double reinforcement reduced the elongation property of the samples with the same geogrid types.

### 3.3. The First Peak Strength and Peak Strength

Figure 7 shows the first peak strength (*f*_1_) values of CTS and CTSG specimens. The *f*_1_ value for CCS is 0.38 MPa, whereas the values for the CTSGs ranged from 0.27 to 0.46 MPa. The geogrid reinforcement has a negligible impact because the *f*_1_ for CTSG depends mainly on the strength of the cement–sand matrix rather than the geogrid-bridging capacity [3,4,5,7,26]. All geogrid types with single and double reinforcement layers can increase the peak strength (*f*_P_) values for CTSG specimens, except CTSG-520-S, due to the insufficient reinforcement layer. The CTSG-560-S and CTSG-580-S contributed *f*_P_ values of 0.52 and 0.59 MPa, whereas CTSG-560-D and CTSG-580-D enhanced the *f*_P_ values of 0.64 and 0.68, respectively. Thus, the *f*_P_ values of the CTSG specimens increased with the increases in geogrid strength and number of reinforcement layers.

### 3.4. Improvement Peak Strength Ratio (ISR)

Figure 8 shows the influence of the geogrid types and the number of reinforcement layers on the ISR values, defined as the *f*_P_ of the CTSG, divided by the *f*_P_ of the CTS. The single and double geogrid layer reinforcements increased the ISR values from 1.00 (no improvement) to 1.21–1.56 and 1.37–1.80, respectively. Both geogrid strength and the number of geogrid layers significantly affect the ISR values. The geogrid strength and the number of geogrid layers strengthen the CTSG specimens by increasing the tensile resistance and preventing crack expansion.

### 3.5. Brittleness Index (BI)

The BI is an index employed to assess the geogrid ability to reduce the brittleness degree of the CTSG specimens [37]. Equation (1) illustrates the definition of the BI for this study. The CTSG specimen exhibiting a BI = 1 behaves like a complete brittle material since the *f*_150_ is zero. The CTSG sample loses a total strength after the peak strength is attained. However, if the BI approaches zero, the CTSG behaves like a ductile material. The CTSG specimen can sustain the *f*_P_ to a deflection of *L*/150 (2 mm) [5].
(1)$$\mathrm{BI}=1-\frac{f_{150}}{f_{P}}$$

Figure 9 illustrates that the reinforcement of all geogrid types could reduce the brittleness compared to the CTS sample (BI = 1). The CTSG-520-S, CTSG-520-D, CTSG-560-S, and CTSG-580-S behaved like a ductile material (BI = 0.00 to 0.11), which falls in the range of CTS reinforced with long polypropylene fibers (BI = 0.01 to 0.04) [5]. In contrast, the CTSG-560-D and CTSG-580-D exhibited semi-brittle (BI = 0.63 to 0.85), similar to the behavior of cemented soil reinforced with steel fibers (BI = 0.60–0.80) [5,37].

The reduction in brittleness mainly depended on the deformability of the specimens [5,37,38]. As discussed in Section 3.1, the CTSG-520-S, CTSG-520-D, CTSG-560-S and CTSG-580-S exhibited flexural response with *f*_P_ ≈ *f*_150_, whereas CTSG-560-S and CTSG-580-S showed flexural response with *f*_P_ < *f*_150_. Thus, the CTSG-520-S, CTSG-520-D, CTSG-560-S and CTSG-580-S have better deformability than the CTSG-560-D and CTSG-580-D. The single reinforcement had more effectiveness than double reinforcement in reducing the brittleness of the CTSG samples. The geogrid strengths had a neglectable impact on the brittleness because the BI values of the CTSG-520-S, CTSG-560-S and CTSG-580-S were almost the same [5,38].

### 3.6. Improvement Toughness Ratio (ITR)

Using CTSG with a high toughness prevents the pavement damage from dynamic loads [25,26]. Thus, comparing the increase in CTSG toughness to the CTS toughness provides valuable details for applying geogrid reinforcement. The ITR is defined as *T*_150_ of CTSG to *T*_150_ of CTS [5]. Figure 10 shows that all geogrid types with single and double reinforcement layers increase the ITR values. This phenomenon indicates that a primary function of the geogrid reinforcement is improving the toughness rather than the peak strength because the ITR values fell to 8.74–11.74, whereas the ISR values were only 1.21–1.80 (Section 3.4). The samples exhibiting high peak strengths also had high ITS values. Consequently, the high ITR values depended on the geogrids’ ability to enhance the peak strength rather than the deformation [5].

### 3.7. Crack Pattern

The crack patterns, such as shape and maximum crack width, of the CTSG specimens were observed because the crack patterns can generally be used as an indicator when characterizing the performance of reinforced materials [3,4,25,26]. The crack pattern of the CTS specimen is a single crack, whereas that of the CTSG is a double crack, as shown in Figure 11. The CTS specimen fails because tensile stress is generated at the bottom of the specimen beam. Thus, the cracks propagate from the bottom to the top. The CTS beam immediately fractured into two parts. It exhibited a conventional brittle failure mode. The single maximum crack width of 30 mm was visually observed due to a concentration of tensile stress near the middle span of the tested beam. This behavior resulted in losing the CTS’ ability to carry residual stresses [3,4,25,26].

In contrast, different crack patterns were observed for the CTSG specimens. After the first crack occurred, the CTSG beam could sustain the residual stress because tensile stress was transferred to the geogrid element, minimizing crack propagation and crack size, as illustrated in Figure 11. The maximum crack widths of the CTSG specimens ranged between 3 and 5 mm, significantly smaller than the CTS specimen. The shear crack propagation is induced by shear stress concentration at the support. The shear crack deviates in the diagonal direction from the support to the location loading application (bearing rod). The small tensile crack subsequently develops at the middle span of the tested beam due to tensile stress. However, only a shear crack is observed for the CTSG-580-D, indicating that a high-strength geogrid with a double reinforcement layer changes the crack pattern from a tensile crack to a shear crack compared to the CCS. Due to the insufficient geogrid strengths of CTSG-520-S, CTSG-560-S, CTSG-520-D, and CTSG-560-S, and the insufficient reinforcement layer of CTSG-580-S, both shear crack, and tensile crack were generated.

### 3.8. Crack Reduction Ratio

Miller and Rifai [28] found that the fiber reinforcement reduced cracks in the desiccated soil by up to 89%, demonstrating the crack resistance of the fiber-reinforcement soils. The crack size decreases as the number of fibers increases. This confirms the effectiveness of fiber reinforcement in inhibiting soil crack from desiccation [29]. Figure 12 presents the crack reduction ratio results, with *R*_cr_ obtained from an image processing technique. All CTSG samples significantly reduce surface cracks, with *R*_cr_ values of 60−89%, close to the crack reduction in fiber-reinforced soils reported by Miller and Rifai [28]. The single and double reinforcement layers show *R*_cr_ = 60−66% and 83−88%, respectively. Thus, the number of reinforcement layers significantly affects the crack reduction rather than the geogrid strength.

### 3.9. Interaction Mechanism of the Fiber-Matrix Interface by a Scanning Electron Microscopy (SEM) Technique

The effect of geogrid reinforcement on the crack patterns and crack widths of CTSG beams, as shown in Figure 11, is present in macro-scale crack behavior, which can be visually observed. However, micro-structural observation is required to investigate micro-scale interaction mechanisms between the interface of the cement-sand matrix and the geogrid surface [4,38]. SEM observation was used to characterize the interfacial bond nature of the geogrid surface. In flexural strength tests, the failure mode is associated with the vertical cracking of the sample caused by the horizontal tensile stresses, as shown in Figure 11, [5]. Before tests, the geogrid surface was bonded with the cement-sand matrix at a curing period of 28 days by chemical and physical adhesions. Figure 13 shows an SEM image of the micro-structure of the uniaxial geogrid surface, cement-sand matrix, hydration products, and various interfaces. The geogrid surface is covered by hydration products that contribute to the bond strength between the geogrid surface and matrix.

Two main interface characteristics of the cement-sand matrix and geogrid were observed, including compact and non-compact interfacial zones. In the compact interfacial zone, the cement-sand matrix around the geogrid surface fully hydrates and fill the space between the geogrid and the matrix, resulting in no observed interface separation [4,5]. Moreover, the hydration product almost entirely covers the surface area of the geogrid, which provides high bond strength at the interface and reduces sliding in the interfacial zone [38]. In contrast, the partial separation between the geogrid surface and matrix was observed in the non-compact interfacial zone [38]. This characteristic is caused by the restriction of the water needed for cement hydration from entering the structure of the cement-sand specimens during the curing period. Therefore, the non-compact interfacial zone exhibits lower high bond strength than the compact interfacial zone.

After the flexural load bent the test specimen, tensile stress was mobilized at the bottom of the test beam. The geogrid was stretched due to the stress transfer mechanism. The shear stress was induced around the interface, creating a bridging ability to sustain the residual flexural stress. As the induced shear stress reached the interfacial bond strength, an enlargement of the crack widths of the test beams tended to increase with continuing applied load [4,5]. However, interfacial bond strength is sufficiently high to present the crack enlargement, minimizing the crack widths. The difference in the crack patterns of the CTSG samples in Figure 11 might be caused by the uniformity of compact and non-compact interfacial zones in each specimen.

## 4. Conclusions

This article presented the effect of uniaxial geogrid strengths and the number of reinforcement layers on the flexural behavior of cement-treated sand reinforced with geogrids. The following conclusions could be drawn following the experimental results:The characteristics of the flexural stress–deflection curves were affected by the uniaxial geogrid strengths and the number of reinforcement layers, which could be classified into main three categories, including (i) deflection-softening behavior with a smaller peak strength than residual strength, (ii) deflection-hardening behavior with a smaller peak strength than residual strength, and (iii) deflection-hardening behavior with a peak strength equal to residual strength.After the first cracks occur, the geogrids start to reinforce the CTSG samples. The high-strength geogrid provided a smaller strain level than the low-strength geogrid. Double reinforcement exhibited a smaller strain value than single reinforcement.The geogrid reinforcement had no impact on the first peak strengths. The peak strengths of the CTSG specimens increased with increasing geogrid strengths and the number of reinforcement layers. The geogrid reinforcement can improve the peak strength of CTS by up to 1.8 times. The geogrid reinforcement improved the toughness rather than peak strength because the improvement toughness ratio is greater than the improvement strength ratio.The crack pattern of the CTS specimen is a single tensile crack, whereas the crack pattern of the CTSG specimen is a double crack, including a shear crack and tensile crack. The single and double reinforcements significantly reduced the surface cracks by an amount ranging from 60% to 88%. The geogrid strength had a negligible impact on the crack reduction.The CTSG is suggested for application as a sub-base layer of pavement and railway structures because the geogrid reinforcement can increase the flexural strength and toughness and reduce surface cracks. Thus, the overall pavement performance and lifetime are improved.

## Figures and Tables

**Figure 1 materials-15-02877-f001:**
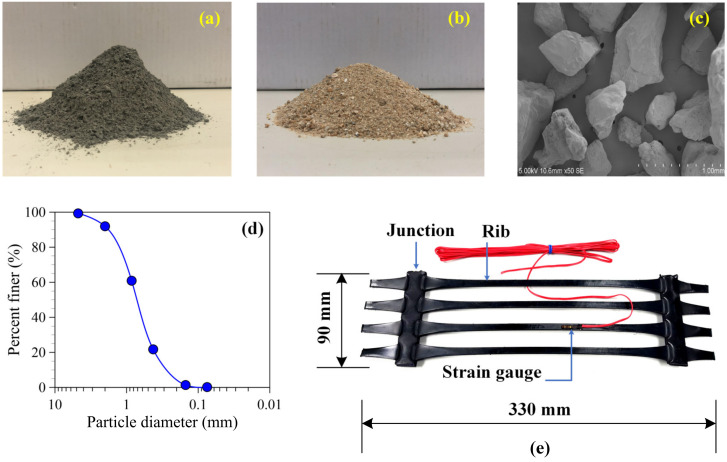
(**a**) cement, (**b**) sand, (**c**) SEM photo and (**d**) particle size distribution curve of sand, and (**e**) uniaxial geogrid used in this study.

**Figure 2 materials-15-02877-f002:**
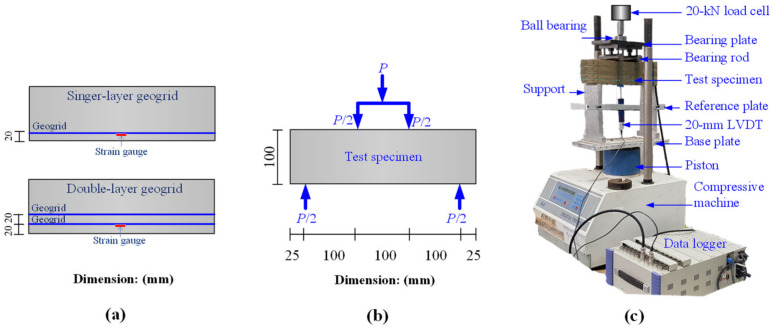
(**a**) reinforcement types; (**b**) test specimen under four-point loading; (**c**) test setup.

**Figure 3 materials-15-02877-f003:**
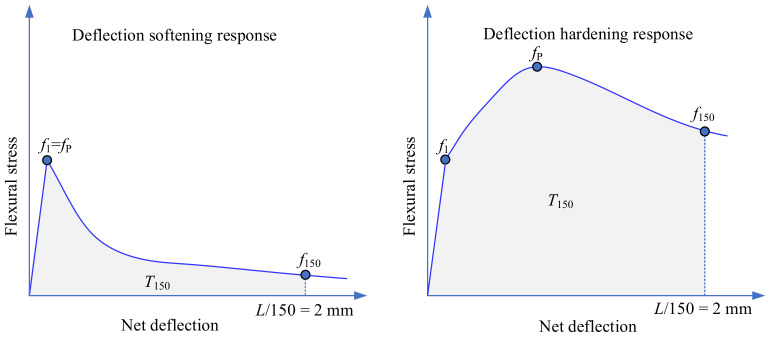
Typical flexural strength-deflection curves.

**Figure 4 materials-15-02877-f004:**
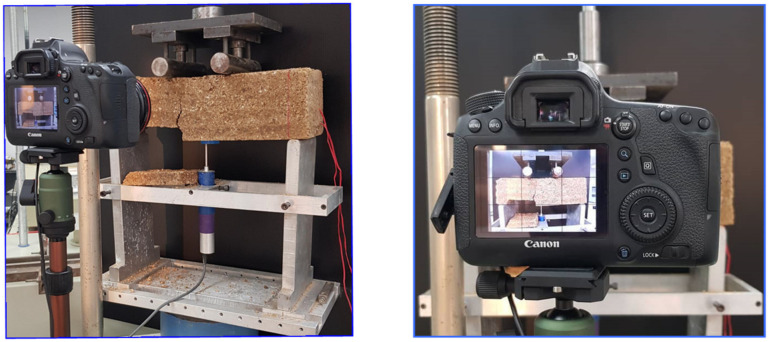
The caption of surface-crack areas of the tested sample used for the image processing technique.

**Figure 5 materials-15-02877-f005:**
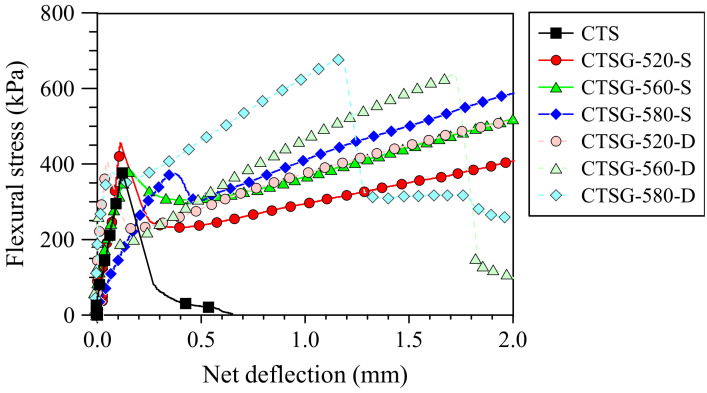
Flexural stress-net deflection curves.

**Figure 6 materials-15-02877-f006:**
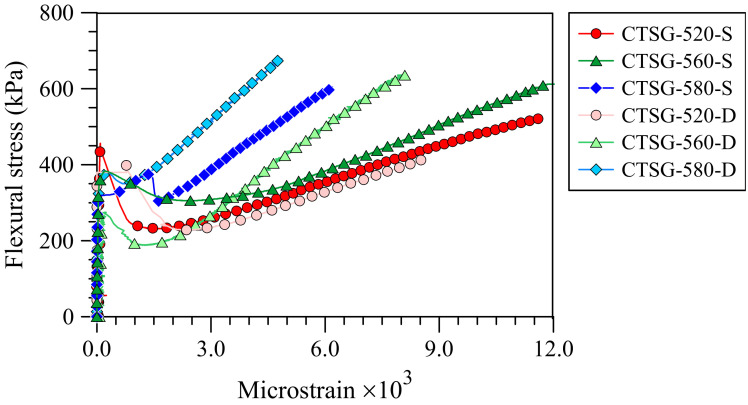
Flexural stress-strain curves.

**Figure 7 materials-15-02877-f007:**
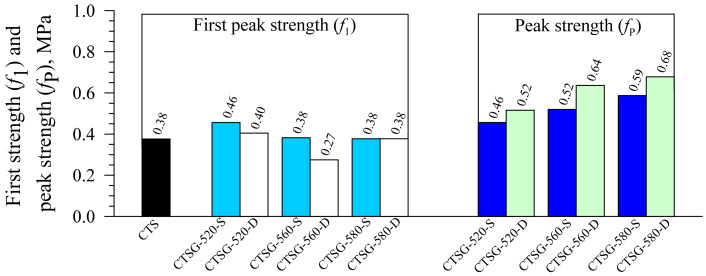
First peak and peak strengths.

**Figure 8 materials-15-02877-f008:**
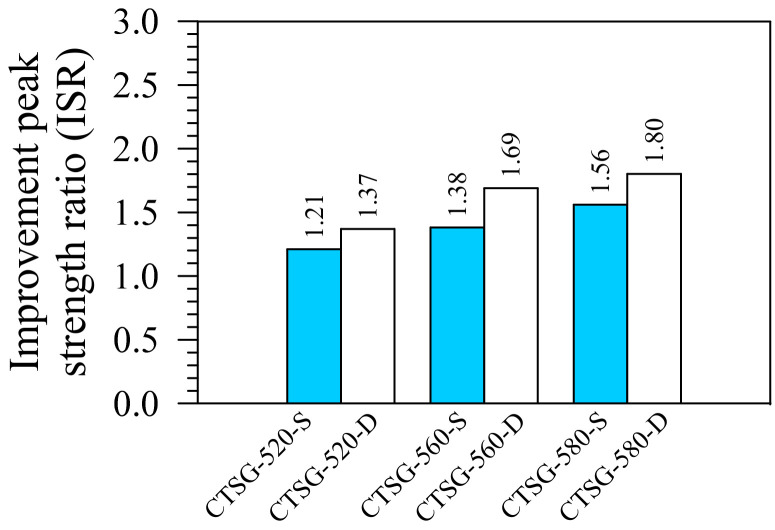
Improvement strength ratios.

**Figure 9 materials-15-02877-f009:**
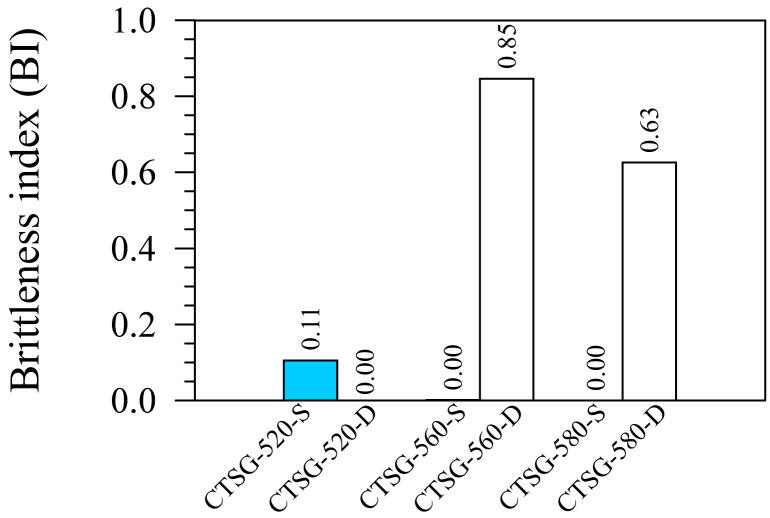
Brittleness indices.

**Figure 10 materials-15-02877-f010:**
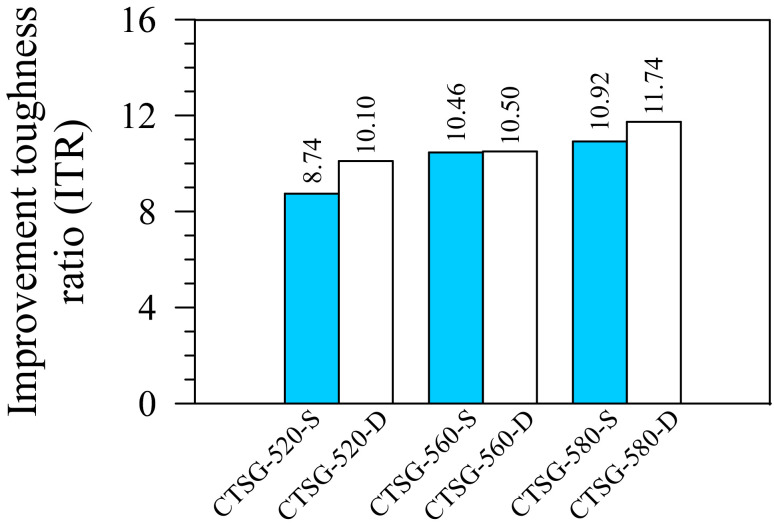
Improvement toughness ratios.

**Figure 11 materials-15-02877-f011:**
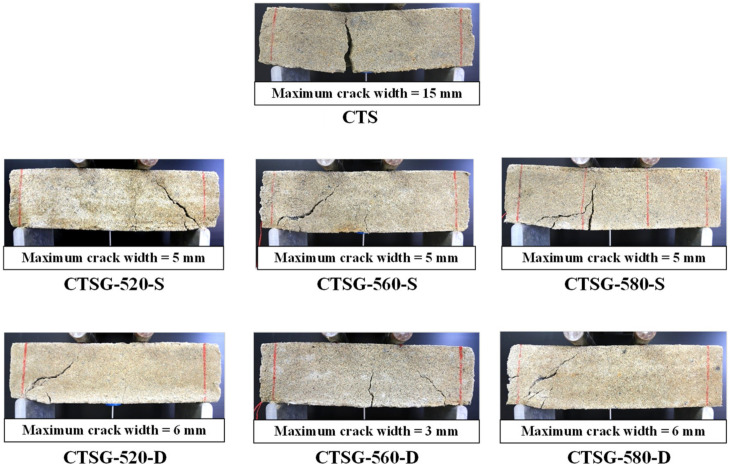
Crack patterns.

**Figure 12 materials-15-02877-f012:**
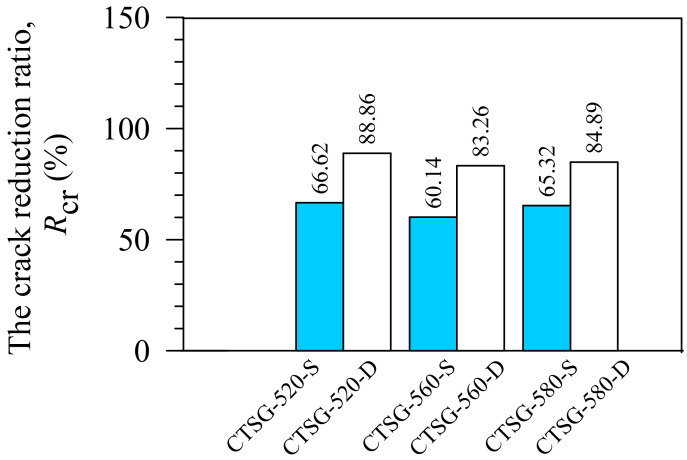
Crack reduction ratios.

**Figure 13 materials-15-02877-f013:**
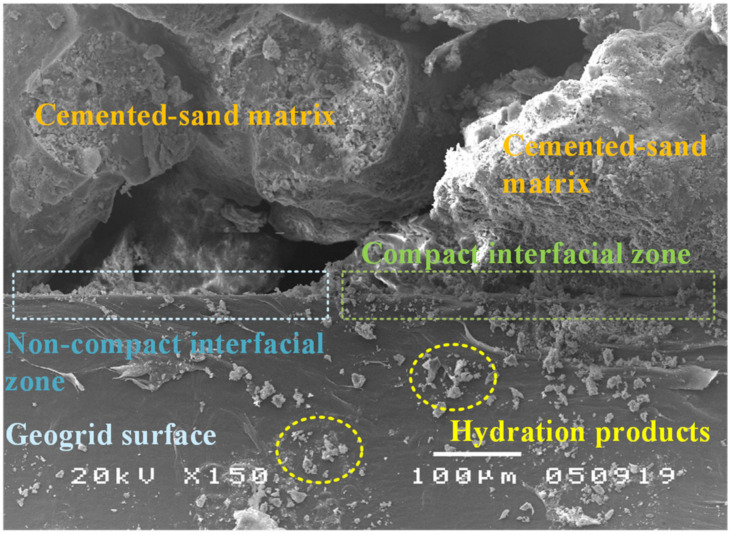
SEM image of the uniaxial geogrid surface in the cement-treated sand.

**Table 1 materials-15-02877-t001:** Geotechnical engineering properties of sand utilized in this study.

Item	Symbol	Value	Unit
Gravel fraction	-	4	%
Sand fraction	-	95	%
Fine fraction	-	1	%
Diameter corresponding to 60% finer	*D* _60_	0.85	mm
Diameter corresponding to 30% finer	*D* _30_	0.50	mm
Diameter corresponding to 10% finer	*D* _10_	0.30	mm
Coefficient of uniformity	*C* _u_	2.83	-
Coefficient of gradation	*C* _c_	0.98	-
Unified soil classification system	USCS	SP	-
Specific gravity	*G* _s_	2.66	-
Maximum dry unit weight	*γ* _d(max)_	18.4	kN/m^3^
Optimum moisture content	*w*	13.8	%

**Table 2 materials-15-02877-t002:** Physical and mechanical properties of the uniaxial geogrid used.

Item	Type of Uniaxial Geogrid			Unit
	RE520	RE560	RE580	
Polymer type	High-Density Polyethylene			
Minimum carbon black content	2			%
Weight per unit area	3.53	6.38	9.61	N/m^2^
Junction strength	95			%
Ultimate strength	53	89	137	kN/m
Strength at 2% strain	13	24	38	kN/m
Strength at 5% strain	25	45	76	kN/m
Strain at ultimate strength	11			%

**Table 3 materials-15-02877-t003:** Sample designation.

Designation	Type of Uniaxial Geogrid	Number of Reinforcement Layers
CTS	-	-
CTS-520-S	RE520	Single layer
CTS-560-S	RE560	Single layer
CTS-580-S	RE560	Single layer
CTS-520-D	RE520	Double layer
CTS-560-D	RE560	Double layer
CTS-580-D	RE560	Double layer

## Data Availability

Not applicable.

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
