# Peer review of "Flexural Performance of Cement-Treated Sand Reinforced with Geogrids for Use as Sub-Bases of Pavement and Railway Structures"

_materials, 2022, doi:10.3390/ma15082877_

Round 1

Reviewer 1 Report

The article by Chuenjaidee et al. entitled "Flexural performance of cement-treated sand reinforced with geogrids for use as subbases of pavement and railway structures" deals with the flexural performance of cement-treated sand reinforced with geogrid (CTSG). The topic is novel, relevant and suitable for publication in the journal, but needs major improvement.

Specific points are listed below:

-The introduction lacks of recent works on the field, which is crucial to show the novelty of the work. See for instance: https://www.sciencedirect.com/science/article/abs/pii/S0266114418300219?via%

-
The abstract is incomplete, does not show summarize the main aims of the work. Should be rewritten.
-
The article is difficult to understand, written in an odd way. Should be rewritten in a more concise way. It will be better to join results and discussion section, otherwise it is difficult to follow for the readers
-
Additional SEM characterization is required to show the failure mode
-
Discussion of the brittleness index is incomplete. Should be compared with literature.
-
Conclusion section is too long and does not summarize the main points of the work.
In
addition, English grammar should be revised through the whole text

Reviewer 2 Report

Thank you for submitting your paper. The work done here draws attention to a significant subject in cement performance for railway pavement and use of geogrids. I have found the paper to be interesting. However, several issues need to be addressed properly especially the language before the paper is being considered for publication. My comments including major and minor concerns are given below:

  1. Please consider reviewing the abstract and highlight the novelty, major findings, and conclusions. I suggest reorganizing the abstract, highlighting the novelties introduced. The abstract should contain answers to the following questions:
  2. What problem was studied and why is it important?
  3. What methods were used?
  4. What conclusions can be drawn from the results? (Please provide specific results and not generic ones).
  5. The abstract must be improved. It does not read well at all. Please use numbers or % terms to clearly shows us the results in your experimental work. Please expand the abstract.
  6. Please consider reporting on studies related to your work from mdpi journals.
  7. The introduction must be expanded, please consider improving the introduction, provide more in-depth critical review about past studies similar to your work, mention what they did and what were their main findings then highlight how does your current study brings new difference to the field.
  8. Lines 41 and 44 avoid bulk citations unless they are given full credit somewhere else in the manuscript.
  9. Table 1 should be referenced unless measured by the authors.
  10. Figures 2 and 3 and 5 add (a) and (b)
  11. Line 134 avoid bulk citations.
  12. Combine sections 3 and 4 into one and call it “Results and Discussion”, each time start with a paragraph talking about the figures below it. Otherwise, I have to keep going up and down to view the figures and then read what you have written about them, its kind of confusing and not realistic.
  13. Line 203 “the flexural stress-deflection curves increase as a linear” does not read well, please rephrase.
  14. Line 204 “Then a sudden decrease in flexural stress is observed.” Why? Don’t just write a sentence like this and then don’t follow it up with an explanation?
  15. Line 206 it does not read well, the sentences are short and do not connect, please check and rephrase.
  16. Discussion section is poorly written, I am unable to understand half of what is written there. This part should be carefully checked and rewritten properly.
  17. Line 206-207 “The flexural stress rapidly reduced to zero. Demonstrates the behavior of specimens without reinforcement weakened after peaking” I have no idea what are the authors are saying here, the sentences are disconnected and do not make sense.
  18. Line 210 same as above…
  19. Line 211 please remove all the bulk citations; I don’t think it is appropriate to have 6 references for a sentence?
  20. Line 212-213, I think I know what is wrong, the authors add dots in their sentences while it should be a continuous sentence with a comma. Please check your writing, at this stage it is not acceptable.
  21. Lines 203-218 combine into one larger paragraph.
  22. Lines 219-246 combine into one larger paragraph.
  23. Lines 231-241 please avoid bullet point style in this part of the article, combine in a paragraph instead and use words such as first, or type 1 or something similar.
  24. Lines 252-253 I am unable to understand this sentence.
  25. The results are merely described and is limited to comparing the experimental observation and describing results. The authors are encouraged to include a more detailed results and discussion section and critically discuss the observations from this investigation with existing literature.
  26. After the authors fix the English, I will be able to provide more comments on the scientific content of the work done. At the moment I am unable to understand many sentences and paragraphs.

Reviewer 3 Report

The manuscript "Flexural performance of cement-treated sand reinforced with geogrids for use as subbases of pavement and railway structures" presents a discussion about the flexural behavior of cement-treated sand reinforced with geogrid (CTSG). The topic is generally relevant and suitable for potential readers, but further revisions are necessary to complement it:

(1) The abstract is not adequate, it lacks a breakdown of the main results and conclusions in quantitative terms, this is important to attract new and potential readers of the manuscript.
(2) The introduction is totally insufficient, important and current topics are not addressed by the authors and there are gaps in the literature that are not addressed, this certainly represents problems in the stage of discussing the results.
(3) The introduction should be improved by inserting new works, the reviewer will suggest 3, but others must be added at the authors' discretion: 10.1016/j.cscm.2021.e00829; 10.3390/fib8110069; 10.3390/ma14216495.
(4) Despite being a potential MDPI standard I do not agree with the full division of the results section with the discussions, they should be merged in my view, if this is not possible, some text and further explanation should be provided. in this section, this is very important.
(5) Some relevant issues such as the use of statistical sampling are not explored, there is not even a line of sampling error by the authors.
(6) Discussion of the Brittleness index (BI) is not adequate, authors should dedicate themselves to improving questions and further exploring this important topic.
(7) Conclusion is extensive, think about reducing and putting the main data in bulleted form.
(8) The English language must be proofread by a native speaker.

Round 2

Reviewer 1 Report

The article has been improved. However, some issues have not been addressed. 

Regarding my point "Additional SEM characterization is required to show the failure mode", what is required is to add more SEM characterization, not to remove the current figures and do not show any SEM analysis. Please carry out a detailed SEM analysis and include it in the revised version of the article. 

Regarding "Discussion of the brittleness index is incomplete. Should be compared with literature". The same as indicated before applies. It is not a matter of removing the brittleness index, it is to compare the current discussion with literature. Please include it in the revised version of the article. 

Reviewer 2 Report

The authors provided the answers to the comments from the first round of review and made sufficient changes in the manuscript according to these comments. I recommend this manuscript for a publication in its present form after the following is met:

1) Remove all bulk citations in the manuscript. this is unacceptable practice unless the authors give full credit to each one of them.

Reviewer 3 Report

All corrections have been made.
